# Comparative Evaluation of the Antimicrobial and Mucus Induction Properties of Selected *Bacillus* Strains against Enterotoxigenic *Escherichia coli*

**DOI:** 10.3390/antibiotics9120849

**Published:** 2020-11-27

**Authors:** Natalia Bravo Santano, Erik Juncker Boll, Lena Catrine Capern, Tomasz Maciej Cieplak, Enver Keleszade, Michal Letek, Adele Costabile

**Affiliations:** 1Department of Life Sciences, University of Roehampton, London SW154JD, UK; nbravosantano@gmail.com (N.B.S.); keleszae@roehampton.ac.uk (E.K.); 2Animal Health Innovation, Chr. Hansen A/S, 2970 Hørsholm, Denmark; dkerbo@chr-hansen.com (E.J.B.); dkleca@chr-hansen.com (L.C.C.); dktoci@chr-hansen.com (T.M.C.); 3Departamento de Biología Molecular, Facultad de Ciencias Biológicas y Ambientales, Universidad de León, 24071 León, Spain; michal.letek@unileon.es

**Keywords:** *Escherichia coli*, adhesion, goblet-cell-derived mucins, probiotics, *Bacillus*

## Abstract

Probiotics have been shown to bind to host receptors, which are important for pathogen adhesion and induce the host’s production of defence factors. They can activate the goblet-cell-derived production of mucins, a major component of the mucus layer and a physical barrier participating in limiting the proximity of microorganisms to the epithelial layer. In the last decade, *Bacillus* spp. strains have gained interest in human and animal health due to their tolerance and stability under gastrointestinal tract conditions. Moreover, *Bacillus* spp. strains can also produce various antimicrobial peptides that can support their use as commercial probiotic supplements and functional foods. The present study aimed to evaluate and determine the ability of selected *Bacillus* spp. strains to inhibit the growth of enterotoxigenic *Escherichia coli* (ETEC) F4 and to reduce binding of ETEC F4 to HT29-16E (mucus-secreting and goblet-like) human intestinal cells. Moreover, mucus production in the HT29 cells in the presence of the *Bacillus* spp. strains was quantified by ELISA. *Bacillus* spp. strains (CHCC 15076, CHCC 15516, CHCC 15541, and CHCC 16872) significantly inhibited the growth of ETEC F4. Moreover, the ability of the probiotic *Bacillus* spp. strains to stimulate mucin release was highly strain dependent. The treatment with *Bacillus subtilis* CHCC 15541 resulted in a significant increase of both MUC2 and MUC3 in HT29-16E cells. Therefore, this strain could be an up-and-coming candidate for developing commercial probiotic supplements to prevent infections caused by ETEC F4 and, potentially, other pathogens.

## 1. Introduction

Gastrointestinal infections are typically treated with antibiotics. However, owing to the increased antimicrobial resistance in recent years and their unfavourable effects on host-microbiota and health, new alternatives have been considered. Due to their characteristics, probiotics are good candidates for preventing and treating gastrointestinal infections and acute diarrhea [1].

To date, much interest and value have been placed on non-sporulated bacteria such as lactobacilli and bifidobacteria as alternatives to antibiotics and as cost-effective treatments for both animal and human gut infections [2]. It is well-known that probiotics may exert direct antagonism by producing antimicrobial compounds, co-aggregating with pathogens, or downregulating their virulent factors [3]. Moreover, through their immune-modulatory actions, probiotics can help prevent or clear the infection more effectively, for example, by inducing the secretion of specific interleukins or promoting the maturation of dendritic cells [4].

The interaction of probiotics with intestinal epithelial cells can also impact the pathogens’ ability to colonize the gastrointestinal tract. Some probiotics have been shown to compete for host cell receptors, which are important for pathogen adhesion and initiation of infection, leading to pathogen exclusion [5]. Moreover, the stimulation of the production of defence factors by the host has also been reported. Probiotics can induce the production of goblet-cell-derived mucins, which are high weight glycoproteins involved in the formation of the mucus layer, a physical barrier participating in limiting the proximity of microorganisms to the epithelial layer [6]. Specific probiotic bacterial strains have been demonstrated to regulate mucin expression, which influences the mucus layer’s properties and indirectly regulates the gut immune system [7,8,9,10]. The increased production of host defensins that may directly kill or inhibit microorganisms has also been investigated [11,12,13].

Several *Bacillus* strains have been screened for their potential probiotic functionalities in several in vitro and in vivo models. Recently, much attention has focused on different *Bacillus* spp. strains in human and animal health due to their tolerance, stability under the gastrointestinal tract conditions, and better stability during heat processing and low temperature storage. In particular, strains of the spore-forming *Bacillus licheniformis*, *Bacillus subtilis*, and *Bacillus amyloliquefaciens* have shown very promising probiotic effects [9,10,14].

Moreover, different *Bacillus* spp. strains seem to possess immune-modulatory and antimicrobial properties [15,16,17]. Furthermore, the use of spore-formers as functional food supplements has been supported due to their significant capacity to produce extracellular enzymes [18]. Nevertheless, it is important to study the mechanism of action of probiotics to reach a consensus about their benefits to human and animal health. This is essential to make them attractive candidates for the development of food, animal, and pharmaceutical products. *Bacillus* spp. are commonly used as probiotic species in the feed industry, however, their benefits need to be confirmed [19,20].

Therefore, this study aims to explore and evaluate the effect of seven selected *Bacillus* spp. strains on pathogenic bacterial growth and adhesion, by focusing on the most recent findings related to their potential impact on human and animal health. To this end, insights on their ability to inhibit the adherence of the enterotoxigenic *Escherichia coli* (ETEC) F4 to the intestinal epithelial HT-29 cell line have been investigated. Moreover, by using mucus-secreting intestinal goblet cells (HT29-16E) from healthy human models, we assessed if these *Bacillus* spp. strains might induce mucin secretion in epithelial cells, which might diminish the binding of enteric pathogens to the mucosal lining.

## 2. Materials and Methods

### 2.1. Bacterial Strains and Culture Conditions

*Bacillus amyloliquefaciens* CHCC 15516, *B. licheniformis* CHCC 3809, and *B. subtilis* CHCC 3810, *B. subtilis* CHCC 9927, *B. subtilis* CHCC 15076, *B. subtilis* CHCC 15541, and *B. subtilis* CHCC 16872 were employed as probiotic bacterial strains. Enterotoxigenic *Escherichia coli* (ETEC) F4 (CHCC 27380) was used as a pathogenic bacterium.

For growth in liquid culture, all *Bacillus* spp. strains and ETEC F4 were cultured in a Luria Bertani broth (LB; Sigma-Aldrich, London, UK). All strains were incubated at 37 °C, with shaking at 200 rpm. *Bacillus* spp. strains were subsequently diluted 1:100 in a LB broth and grown with shaking for 3 h to an early exponential phase. For growth on a solid medium, *Bacillus* strains were grown on Trypticase Soya Agar (TSA; Sigma-Aldrich, London, UK) plates and incubated at 30 °C, whereas pathogenic bacteria were grown on LB agar plates and incubated at 37 °C. For the agar well diffusion assay, ETEC F4 was grown on TSA plates, and incubated at 37 °C [21].

### 2.2. Cell Lines and Culture Conditions

HT29-19A (non-mucus secreting) and HT29-16E (mucus-secreting) clones [22] were cultured in 100 mm cell culture plates (Sarstedt) with Dulbecco’s Modified Eagle’s medium (DMEM, Sigma-Aldrich, London, UK) containing pyruvate, glucose, and glutamine and supplemented with a 10% heat-inactivated foetal bovine serum (FBS; Sigma-Aldrich) and 5% of the penicillin and streptomycin solution (Sigma-Aldrich, London, UK) unless otherwise specified. Cell lines were maintained in a 5% CO_2_ incubator at 37 °C and all the experiments were carried out in between passages 5 and 10 [23].

### 2.3. Pathogen Inhibition (Agar Well Diffusion) Assay

The day before the experiment, *Bacillus* spp. strains and ETEC F4 were grown at 37 °C with shaking for 18 h. On the day of the experiment, LB agar (LB; Sigma-Aldrich, London, UK) was melted and cooled to 40–45 °C. ETEC F4 was normalized to 0.5 McFarland and 10 µL was added to 30 mL of LB agar (~3 × 10^4^ CFU/mL) and mixed. The agar was then transferred to an omnitray (Thermo Scientific, Waltham, MA, USA, 242811) and a 96-well Immuno TSP lid (Thermo Scientific, 445497) was placed on top. Plates were left to solidify and dry for 45 min. A total of 5 µL overnight culture of the probiotic strains (~5 × 10^5^ CFU/mL) or 2 µL ciprofloxacin (Sigma-Aldrich, London, UK) (~0.02 µg/mL) was then added to the wells. Following the 24 h incubation at 30 °C, the plates were scanned, and the inhibition zones were measured diagonally or from mid-well to full growth in duplicate. The well diffusion assay was performed following the guidelines of the Clinical and Laboratory Standards Institute (CLSI manual for antimicrobial susceptibility, 2018) [24].

### 2.4. Adhesion Assay

HT29-19A and HT29-16E cells were seeded in 24-well plates in complete DMEM (supplemented with penicillin and streptomycin) at a cell density of 1.5 × 10^5^ and 1.2 × 10^5^, respectively, and incubated for 5–7 days, respectively, until confluency was reached.

The adhesion assay was performed using the principle method described by Tian et al. [25]. On the day of the experiment, *Bacillus* spp. strains and ETEC F4 grown as previously described were washed and resuspended in DMEM without antibiotics and normalized to an optical density at 600 nm (OD_600_) of 3.8 and 0.5, respectively. The cell monolayers were gently washed once with PBS, after which 1 mL of antibiotic-free DMEM containing OD_600_-normalized probiotic strains and diluted 1:10 were added to the wells (~3 × 10^7^ CFU per well; a multiplicity of infection (MOI) ~10). The untreated wells were included as negative controls.

After 1 h of incubation at 37 °C, 100 µL of normalized ETEC F4 was added to each well (~3 × 10^7^ CFU per well; MOI = 10) and returned to the incubator for another 2 h.

Lastly, the medium was aspirated, cells were washed three times with PBS and lysed with 0.1% Triton X-100. Cell lysates were resuspended in 1 ml of PBS and serial dilutions (10^−3^, 10^−4^, and 10^−5^) were plated onto MacConkey agar plates to quantify the amount of adhering ETEC F4 by colony forming units (CFU) counting.

### 2.5. MUC2 and MUC3 Quantification

HT29-19A and HT29-16E cells were cultured in 24-well plates as described above. HT29-16E cell monolayers were gently washed once with PBS after which 1 mL of antibiotic-free DMEM containing OD_600_-normalized probiotic strains (grown as previously described) and diluted 1:10 were added to the wells (including untreated wells used as negative controls) as mentioned above. Then, the cells were incubated at 37 °C until the desired time point was reached.

Secreted MUC2 and membrane-bound MUC3 levels were quantified by ELISA assays as previously described [26]. MUC2 (Abbexa, UK, cat abx055282) and MUC3 (Abbexa, UK, cat abx152398) ELISA kits were used to determine the level of these mucins in supernatants and cell lysates, respectively.

First, HT29-16E cell supernatants were transferred to sterile tubes. The cells were then washed once with PBS to recover all non-adherent mucins. The mix of cell supernatants and recovered PBS was centrifuged at 2500× g for 10 min to remove any precipitates. The MUC2 production was measured in the resulting supernatant.

In parallel, HT29-16E cells were detached with 0.25% trypsin and collected by centrifugation at 1800× g for 10 min to remove the supernatant. Afterward, cells were washed three times in ice-cold PBS, lysed by ultrasonication, and centrifuged to remove all cell debris. The MUC3 ELISA assay was carried out with the resulting supernatant.

### 2.6. Statistical Analyses

Graph plotting and statistical tests were conducted using the GraphPad Prism software (version 7.0, Inc., San Diego, CA, USA). Significant differences across treatments were assessed by running either One-way or Two-way ANOVA followed by post hoc multiple comparison tests. A *p*-value ≤ 0.05 was considered statistically significant.

## 3. Results and Discussion

### 3.1. Bacillus Subtilis and Bacillus Amyloliquefaciens Strains Reduce the Growth of Pathogenic ETEC F4

We first screened for the direct antimicrobial activity of our collection of seven *Bacillus* spp. strains: One *Bacillus amyloliquefaciens* (CHCC 15516), one *Bacillus licheniformis* (CHCC 3809), and five *Bacillus subtilis* strains (CHCC 3810, CHCC 9927, CHCC15076, CHCC 15541, and CHCC 16872). To this end, we employed an agar well diffusion-based method to assess the inhibitory effect of the probiotic strains against an enteric pathogen: Enterotoxigenic *Escherichia coli* (ETEC) F4 (Figure 1).

We observed significant changes after treatments with different probiotic strains. CHCC 3809, CHCC 3810, and CHCC 9927 all failed to inhibit the growth of ETEC F4. In contrast, the remaining four *Bacillus* spp. strains significantly inhibited the growth of ETEC F4, with three *Bacillus subtilis* strains CHCC 15076, CHCC 15541, and CHCC 16872 exhibiting the most profound inhibitory effect.

### 3.2. Bacillus spp. Strains Reduce Pathogenic ETEC F4 Binding to HT29 Cells

Toxin delivery by ETEC requires a direct engagement of intestinal cells [27]. However, this is significantly hindered by intestinal mucins such as mucin 2 (MUC2), the major secreted mucin in the intestinal lumen [28], or by membrane-bound mucins such as mucin 3 (MUC3) [29].

Therefore, we performed an exclusion adhesion screening of the seven *Bacillus* spp. strains with HT29-19A (non-mucus-producing) and HT29-16E (mucus-producing) cells. Our working hypothesis was that any inhibition of ETEC adhesion mediated by *Bacillus* spp. might be due to the induction of mucus production in intestinal cells. Therefore, HT29-19A and HT29-16E cells were treated with the seven selected *Bacillus* spp. strains, and the ETEC F4 strain was used as the pathogenic bacterium.

Importantly, all *Bacillus* spp. strains used in this study caused a significant reduction of ETEC F4 adhesion to both HT29-19A and HT29-16E (Figure 2) cell monolayers, with slight differences among bacterial strains in both cell lines. In HT29-19A cells, the highest and most significant reduction of ETEC adhesion was observed after treatment with CHCC 15076, CHCC 15541 (both *Bacillus subtilis*), and CHCC15516 (*Bacillus amyloliquefaciens*). By contrast, treatments with CHCC 3809 (*Bacillus licheniformis*) and CHCC 3810 (*Bacillus subtilis*) caused the most significant drop of ETEC F4 adhesion to HT29-16E cells (Figure 2).

In addition, ETEC adhesion to intestinal cells was equivalent when we compared HT29-19A and the mucus-producing HT29-16E cells (Figure 3). Overall, our results suggest that all *Bacillus* spp. strains used in this study can inhibit the adhesion of ETEC to intestinal cells, but this effect was not due to the induction of mucus production at short periods of incubation.

### 3.3. Bacillus Subtilis and Bacillus licheniformis Strains Increase Mucin Production in HT29-16E Cells at Long Periods of Incubation

Our ETEC F4 adhesion exclusion screening results were very promising but partially unexpected regarding the influence of mucus production on the inhibition of pathogen adhesion to intestinal cells. ETEC secretes metalloproteases to degrade mucin 2 and 3 actively and facilitate its adhesion to intestinal epithelial cells [30]. Therefore, we expected profound differences when HT29-19A (non-mucus-producing) and HT29-16E (mucus-producing) findings were compared.

We hypothesized that the incubation time used in our screening with different *Bacillus* spp. strains (1 h) was too short to have any impact on mucin production. Therefore, we analyzed the mucin production in HT29-16E cell monolayers by ELISA following different time periods of incubation with *B. subtilis* CHCC 15541, one of the most promising candidates based on the two previous assays. MUC2 is secreted to the medium, and it was therefore quantified from the cells’ supernatant. Whereas, MUC3 is a membrane-bound mucin, and it was extracted from the cell lysates [31]. We observed a trending increase in mucin production at 10 h (Appendix A
Appendix A).

Then, we quantified MUC2 and MUC3 production by ELISA in HT29 cells after a 10-h treatment with the seven selected *Bacillus* spp. strains (Figure 4). For both mucins, significant differences were observed after treating HT29-16E cell monolayers with different bacterial strains. In particular, treatment with *Bacillus subtilis* CHCC 15541 resulted in a significant increase of both MUC2 and MUC3 in HT29-16E cells (Figure 4).

Interestingly, the ability of the probiotic *Bacillus* sp. strains to stimulate mucin release was highly strain-dependent. For example, while the treatment of HT29-16E cells with *Bacillus subtilis* CHCC 15076 resulted in a significant increase of mucin 2, no significant differences were found in mucin 3 production between untreated and CHCC 15076-treated cells (Figure 4). In contrast, the treatment of HT29-16E cells with *Bacillus licheniformis* CHCC 3809 showed a significant increase in mucin 3 production, whereas, the effect of CHCC 3809 treatment on mucin 2 production in HT29-16E cells was not statistically significant when compared to the untreated cells (Figure 4).

In summary, our data suggested that some of the *Bacillus* spp. strains tested in this study may exhibit an antimicrobial activity against pathogenic bacteria, as examined here for ETEC. Indeed, the bacteria of the genus *Bacillus* are very well-known producers of different antimicrobial compounds, including bacteriocins and peptide or lipopeptide antibiotics [32].

Alternatively, all of the *Bacillus* spp. strains may inhibit an ETEC attachment to intestinal cells by an unknown mechanism of action, which is functional even at very short periods of incubation, when mucus production is not relevant. This could be due to the competitive exclusion of pathogenic microorganisms or other yet unknown probiotic-mediated disruptions of host-pathogen interactions [33]. Similar to what has been described in other models of infection, Bacillus spp. may block the host cell receptors on the surface of the gastrointestinal tract, which are essential for the adhesion of different pathogens to enteric cells [5].

Finally, some of the strains tested may induce the production of both secreted and membrane-bound mucins after long periods of incubation, and this effect is strain specific. Interestingly, short-chain fatty acids produced by intestinal microbiota, such as butyrate, may stimulate mucin production in intestinal cells, particularly MUC2 [34,35].

Remarkably, *Bacillus subtilis* CHCC 15541 is the only strain that showed a positive effect in all assays. Therefore, this strain is an up-and-coming candidate for developing commercial probiotic supplements and functional foods to prevent infections with ETEC and potentially other pathogens.

## 4. Conclusions

The antimicrobial resistance has become a global threat within the past decades, and such a threat has been accelerated due to the current COVID-19 pandemic, since many hospitalized patients have received broad-spectrum antibiotics to prevent secondary bacterial infections [36]. The development of novel strategies to overcome antimicrobial resistance is now extremely urgent, especially for Gram-negative pathogens such as pathogenic *E. coli*. Enterobacteria are already considered the number one priority by WHO regarding the research and development of new antibiotics [37]. Given the few therapeutic options available for treating infections caused by multi-drug resistant *E. coli*, the most promising anti-infectives are developed for human clinical medicine. However, many of these strains are of animal origin, which facilitates their acquisition of antimicrobial resistance traits [38]. Therefore, urgent preventative measures are needed to complement traditional antibiotherapy in both human and animal populations.

One solution is the use of probiotics, which have been actively researched over the past few decades. An increasing interest on functional foods has been focused on the hypothesis that specific probiotics may modulate and interact with the gut barrier, re-establishing gut homeostasis. However, a lack of knowledge on their mechanism of action hampers the development of efficient preventative tools based on these microorganisms. Here, we screened seven very promising *Bacillus* spp. strains and analyzed their inhibition of growth of ETEC on agar plates, as well as their inhibition of ETEC adhesion to intestinal cells. Furthermore, we analyzed the role of mucus induction on this beneficial effect for gut health. Our results suggest that *Bacillus* spp., in particular *B. subtilis* CHCC 15541, might be well-armed to control the adhesion of pathogens in the intestine and that different molecular factors mediate this [39,40]. This research work may facilitate the bioengineering of *Bacillus* spp. to produce highly effective probiotic strains in the future.

## Figures and Tables

**Figure 1 antibiotics-09-00849-f001:**
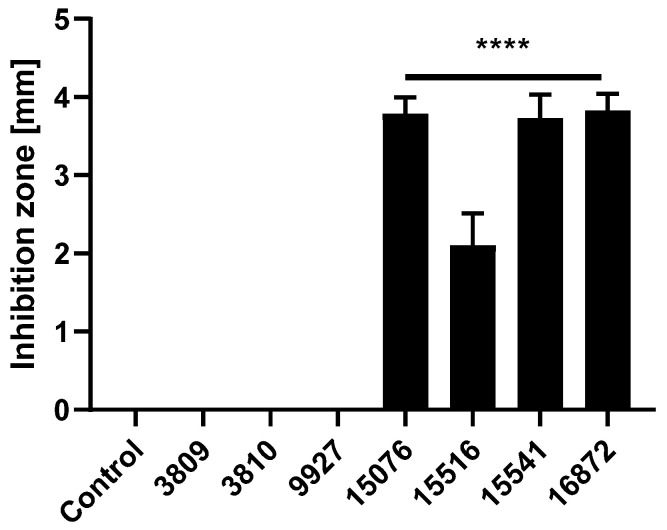
Agar well diffusion-based inhibition of pathogenic growth by seven *Bacillus* spp. strains. Seven *Bacillus* spp. strains (CHCC 3809, CHCC 3810, CHCC 9927, CHCC 15076, CHCC 15516, CHCC 15541, and CHCC 16872) were spotted on agar plates casted with *Escherichia coli* (ETEC) F4. Inhibition zones were measured after 24 h of incubation. Data are depicted as the width of the well (3.5 mm) subtracted from the diameter of the inhibition zone. Data are expressed as mean ± standard deviation (SD) of two experiments performed in triplicates or quadruplicates. One-way ANOVA and multiple comparison Dunnett’s tests were performed to assess statistically significant differences between the negative control group (with no inhibition zone) and other probiotic treatments. *p*-value ≤ 0.0001 (****).

**Figure 2 antibiotics-09-00849-f002:**
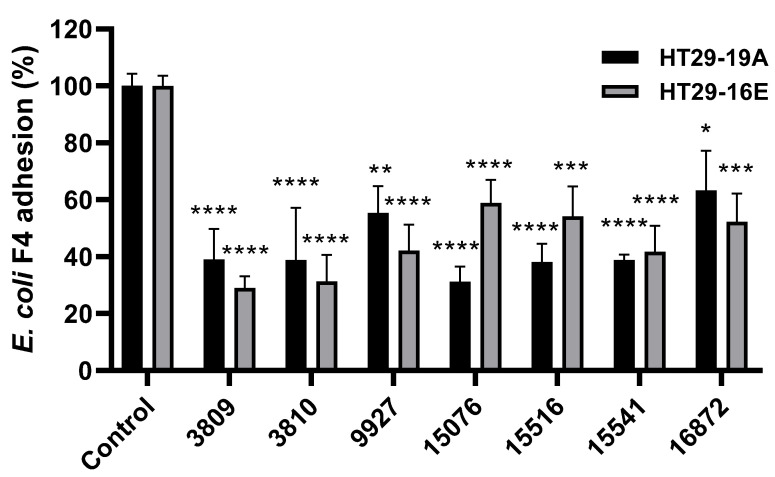
Adhesion of ETEC F4 to HT29-19A and HT29-16E cells after pre-treatment with seven *Bacillus* spp. strains. HT29-19A and HT29-16E cell monolayers were incubated with the seven *Bacillus* spp. strains for 1 h before adding ETEC F4, and then incubated for an additional 2 h. Adherent ETEC F4 was quantified by colony forming units (CFU) counting on MacConkey agar plates. The percentage of adhesion was calculated by dividing the number of adhering ETEC F4 with the inoculum. The depicted data are normalized by the amount of adhering ETEC F4 counted in the control group. Data are expressed as mean ± standard error of the mean (SEM) of nine experiments performed in triplicates. One-way ANOVA and multiple comparison Tukey’s tests were performed to assess statistically significant differences between control and different probiotic treatments for each cell line. *p*-value ≤ 0.05 (*); ≤0.01 (**); ≤0.001 (***); ≤0.0001 (****).

**Figure 3 antibiotics-09-00849-f003:**
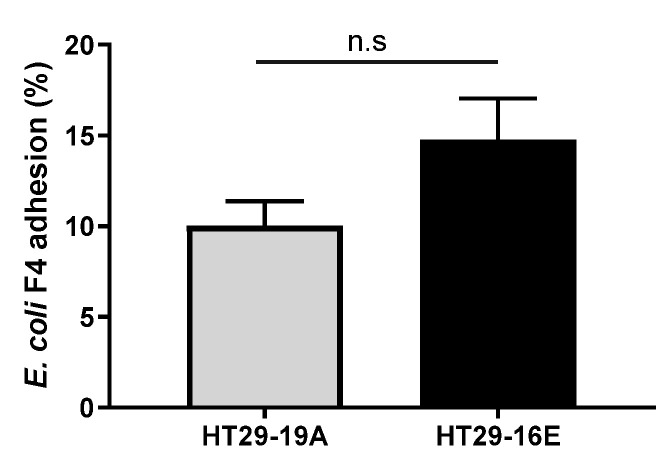
*E. coli* F4 adhesion assay with HT29-19A and HT29-16E cells. HT29-19A and HT29-16E cell monolayers were incubated with Dulbecco’s Modified Eagle’s medium (DMEM) without antibiotics for 1 h before adding *E. coli* F4, and then incubated for an additional 2 h. Adherent *E. coli* F4 was quantified by CFU counting on MacConkey agar plates, and the percentage of adhesion was normalized by initial *E. coli* F4 bacterial load. Data are expressed as mean ± SEM of nine experiments performed in triplicates; n.s.: not significant. A student t-test was performed to assess statistically significant differences between both cell lines.

**Figure 4 antibiotics-09-00849-f004:**
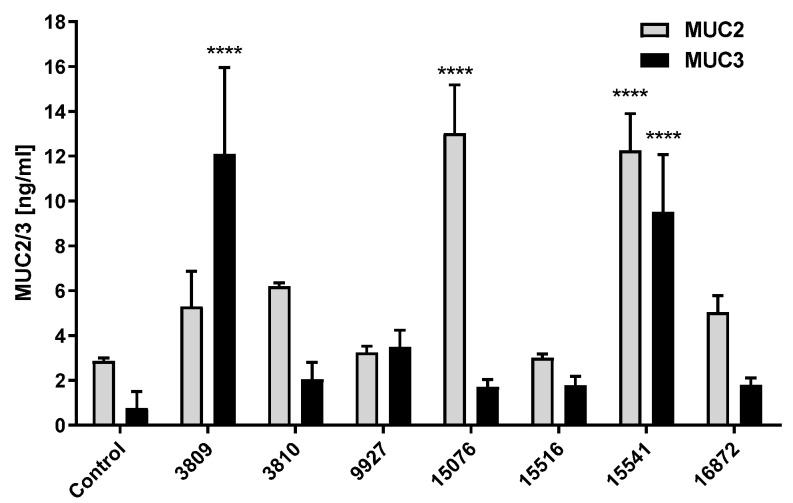
Mucin 2 and mucin 3 quantification in HT29-16E cells after treatment with seven *Bacillus* spp. strains. HT29-16E cells were incubated with the seven *Bacillus* spp. strains for 10 h. The cells’ supernatants were taken and mucin 2 (MUC2) production was measured in this fraction by ELISA. The cells were then detached with trypsin, collected and lysed by ultrasonication, and mucin 3 (MUC3) was measured in this cell fraction by ELISA. Data are expressed as mean ± SEM of two independent experiments performed in duplicates. Two-way ANOVA and multiple comparison tests were performed to assess statistically significant differences between control and different probiotic treatments for MUC2 and MUC3, respectively. *p*-value ≤ 0.0001 (****).

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
