# Peer review of "Comparative Evaluation of the Antimicrobial and Mucus Induction Properties of Selected Bacillus Strains against Enterotoxigenic Escherichia coli"

_antibiotics, 2020, doi:10.3390/antibiotics9120849_

Round 1

Reviewer 1 Report

In this manuscript, the antimicrobial and mucus induction properties of selected Bacillus strains against enterotoxigenic Escherichia coli (ETEC) F4 are discussed. The ideas described seem to be worth studying, but comments below should be addressed before publication can be considered.

  1. Studies have shown that Bacillus spp. strains can significantly reduce the adhesion of ETEC F4 to HT29 (goblet-like) cells. Does it have the same effect on other intestinal epithelial cells such as Paneth cells and enteroendocrine cells?
  2. Whether the increase in mucus and mucin secreted by HT29 cells is directly related to the decrease in ETEC F4 adhesion may need to be mentioned.
  3. These results provide new insights into the protective and anti-infective effects of probiotics on gut health and is recommended for acceptance.

Author Response

Journal: Antibiotics (ISSN 2079-6382)

Manuscript ID: antibiotics-1012202

Type: Communication

Number of Pages: 11

Title: Comparative evaluation of the antimicrobial and mucus induction properties of selected Bacillus strains against enterotoxigenic Escherichia coli

Authors: Natalia Bravo Santano, Erik Juncker Boll, Lena Catrine Capern, Tomasz Maciej Cieplak, Enver Keleszade, Michal Letek and Adele Costabile

Comments of the Reviewers and Editor are in black, responses are in blue and citations from and changes in the manuscript are in red.

Reviewer 1 - Review Report (Round 1)

Comments and Suggestions for Authors

In this manuscript, the antimicrobial and mucus induction properties of selected Bacillus strains against enterotoxigenic Escherichia coli (ETEC) F4 are discussed. The ideas described seem to be worth studying, but comments below should be addressed before publication can be considered.

  1. Studies have shown that Bacillus spp. strains can significantly reduce the adhesion of ETEC F4 to HT29 (goblet-like) cells. Does it have the same effect on other intestinal epithelial cells such as Paneth cells and enteroendocrine cells?

We have not tested this hypothesis, which is very interesting. We have preliminary unpublished data showing that the same Bacillus sp. strains reduce binding of an E. coli O157 strain to Caco-2 human intestinal epithelial cells. However, these data are not answering your question, and since they are based on a different bacterial strain, we did not consider them of relevance here. It would be interesting to test this effect on other intestinal epithelial cells, but we believe this is out of the scope of this short communication.

  1. Whether the increase in mucus and mucin secreted by HT29 cells is directly related to the decrease in ETEC F4 adhesion may need to be mentioned.

Yes, we have addressed this in lines 180-182.

  1. These results provide new insights into the protective and anti-infective effects of probiotics on gut health and is recommended for acceptance.

Many thanks for your very positive revision.

Reviewer 2 Report

The authors describe an interesting preliminary study evaluating potential probiotic effects of different spore-forming strains of Bacillus spp. on gut health, in terms of ability of selected strains to inhibit the growth of E. coli F4 and to reduce its binding to mucus-secreting and goblet-like human intestinal cells.

Methodology is well applied to achieve the proposed objectives. The paper is generally well-written and, as mentioned by the authors, results obtained may provide new insights into the beneficial effects of probiotics on gut health. However, further both in vitro and in vivo experiments are required to produce highly effective and safe probiotics strains to use them as therapeutic agent.

From my point of view, minor changes must be performed before final publication.

ABSTRACT should be substantially reviewed by authors in order to provide key information about methods used in this study (experimental groups and treatments tested). Moreover, both results and conclusions are described in general terms, and specific results should be described, mainly promising probiotic effect of Bacillus subtilis CHCC 15541. Finally, background should be summarized.    

Authors include receptors as keyword, but neither receptor-binding assay is performed. Thus, receptors could be eliminated from keywords section.

INTRODUCTION supports background information necessary to provide a specific context for the results, explaining both therapeutic effects of probiotics in order to prevent and treat gastrointestinal infections and their potential mechanisms of actions. However, references about probiotic effects of spore-forming strains of Bacillus should be included (lines 57-60). Moreover, authors should explain why they use Bacillus spp. strains mentioned in methods section. Are there any reasons or preliminary studies for it? Objectives are clearly defined.

MATERIALS AND METHODS are clearly explained, providing detailed information about both bacterial strains and cell lines culture conditions, pathogen inhibition assay, adhesion assay and mucins quantification. However, the information provides is not referenced. This should be mandatory in assays performed, indicating if experimental conditions are originals or modifications of established protocols.

In statistical analyses, authors should indicate what p-value is considered statistically significant (I understand that p-value <0.05 is statistically significant).

Respect to Results and Discussion section, results are generally clear, and their description is in accordance with the graphics. Both objectives and key points of experimental procedures of each analysis are included in this section, providing valuable information about the steps performed as well as issues raised during the study.

Data about mucin production at different time periods of incubation with selected strains should be included as supplemental material.

Authors support that “no difference was observed in the control group between HT29-19A and HT29-16E cell monolayers (data not shown) (lines 223-225). However, this result is included in figure 4. Can you confirm if this correct? 

Discussion on results obtained ranges from lines 246 to 261. However, due to manuscript format, discussion of specific results should be included in its respective section, and not at the end of section. Due to its importance in manuscript, authors should widely discuss alternative mechanism of action at short periods of incubation (lines 250-254).

Are there in vitro or in vivo studies that also support the importance of Bacillus subtilis CHCC 15541 as probiotic?

Authors include final conclusions about their study and its clinical importance. However, conclusions are broad, and authors should also include specific conclusion about the potential probiotic effect of B. subtilis 1554 (it is the main finding obtained from this study).

Author Response

Journal: Antibiotics (ISSN 2079-6382)

Manuscript ID: antibiotics-1012202

Type: Communication

Number of Pages: 11

Title: Comparative evaluation of the antimicrobial and mucus induction properties of selected Bacillus strains against enterotoxigenic Escherichia coli

Authors: Natalia Bravo Santano, Erik Juncker Boll, Lena Catrine Capern, Tomasz Maciej Cieplak, Enver Keleszade, Michal Letek and Adele Costabile

Comments of the Reviewers and Editor are in black, responses are in blue and citations from and changes in the manuscript are in red.

Reviewer 2 - Review Report (Round 1)

Comments and Suggestions for Authors

The authors describe an interesting preliminary study evaluating potential probiotic effects of different spore-forming strains of Bacillus spp. on gut health, in terms of ability of selected strains to inhibit the growth of E. coli F4 and to reduce its binding to mucus-secreting and goblet-like human intestinal cells.

Methodology is well applied to achieve the proposed objectives. The paper is generally well-written and, as mentioned by the authors, results obtained may provide new insights into the beneficial effects of probiotics on gut health. However, further both in vitro and in vivo experiments are required to produce highly effective and safe probiotics strains to use them as therapeutic agent.

From my point of view, minor changes must be performed before final publication.

ABSTRACT should be substantially reviewed by authors in order to provide key information about methods used in this study (experimental groups and treatments tested). Moreover, both results and conclusions are described in general terms, and specific results should be described, mainly promising probiotic effect of Bacillus subtilis CHCC 15541. Finally, background should be summarized.

We thank the Reviewer for this point, and now the abstract has been revised by making it less general and more specific in line with the data obtained.

Authors include receptors as keyword, but neither receptor-binding assay is performed. Thus, receptors could be eliminated from keywords section.

“Receptors” has been removed as a keyword accordingly.

INTRODUCTION supports background information necessary to provide a specific context for the results, explaining both therapeutic effects of probiotics in order to prevent and treat gastrointestinal infections and their potential mechanisms of actions. However, references about probiotic effects of spore-forming strains of Bacillus should be included (lines 57-60).

We thank the Reviewer for this point. We added a new sentence and references as follow:

Adibpour et al. (2019). Applied Food Biotechnology, 6(2), 91-100.

Celandroni et al. (2019) PLOS ONE 14(5): e0217021.

Elshaghabee et al. (2017) Front Microbiol. 2017; 8:1490. pmid:28848511.

Moreover, authors should explain why they use Bacillus spp. strains mentioned in methods section. Are there any reasons or preliminary studies for it? Objectives are clearly defined.

Thanks for this comment; we have used a random sample of our collection of probiotic strains.

MATERIALS AND METHODS are clearly explained, providing detailed information about both bacterial strains and cell lines culture conditions, pathogen inhibition assay, adhesion assay and mucins quantification. However, the information provides is not referenced. This should be mandatory in assays performed, indicating if experimental conditions are originals or modifications of established protocols.

We thank the Reviewer for this point. It has been revised accordingly, and new references have been added to the manuscript (Ref. 21-26)

In statistical analyses, authors should indicate what p-value is considered statistically significant (I understand that p-value <0.05 is statistically significant)

We have now added this information and revised accordingly.

Respect to Results and Discussion section, results are generally clear, and their description is in accordance with the graphics. Both objectives and key points of experimental procedures of each analysis are included in this section, providing valuable information about the steps performed as well as issues raised during the study.

Data about mucin production at different time periods of incubation with selected strains should be included as supplemental material.

Thanks, this is now Figure S1.

Authors support that “no difference was observed in the control group between HT29-19A and HT29-16E cell monolayers (data not shown) (lines 223-225). However, this result is included in figure 4. Can you confirm if this correct?

Many thanks for raising this point. No, this dataset is not included in Figure 4; therefore, we removed this sentence to avoid any further confusion.

Discussion on results obtained ranges from lines 246 to 261. However, due to manuscript format, discussion of specific results should be included in its respective section, and not at the end of section. Due to its importance in manuscript, authors should widely discuss alternative mechanism of action at short periods of incubation (lines 250-254).

Many thanks for raising this point. We have merged the results and discussion to improve the readability of this communication. With this aim in mind, we have summarized the manuscript's findings at the end of this section. Regarding mechanisms of action at short periods of incubation, we have expanded our discussion towards the possibility that Bacillus spp. could block the adherence of pathogens by competition for host cell-binding sites.

Are there in vitro or in vivo studies that also support the importance of Bacillus subtilis CHCC 15541 as probiotic?

In vitro models of the human stomach and in vivo studies with probiotic Bacillus spp. strains have been performed to understand better all biochemical effects such as antimicrobial and enzymatic activity, thus protecting GIT and other infections.

Bacillus subtilis is commonly used as probiotic species in the feed industry. Recent studies have described a high throughput screening combined with the detailed characterization of endospore-forming bacteria to identify new Bacillus spp. strains for use as probiotic additives in animal feed. This point has been now added in the introduction with new references:

Larsen, N., Thorsen, L., Kpikpi, E.N. et al. Characterization of Bacillus spp. strains for use as probiotic additives in pig feed. Appl Microbiol Biotechnol 98, 1105–1118 (2014). https://doi.org/10.1007/s00253-013-5343-6

Jeżewska-Frąckowiak J, Seroczyńska K, Banaszczyk J, Jedrzejczak G, Żylicz-Stachula A, Skowron PM. The promises and risks of probiotic Bacillus species. Acta Biochim Pol. 2018 Dec 6;65(4):509-519. doi: 10.18388/abp.2018_2652. PMID: 30521647

Authors include final conclusions about their study and its clinical importance. However, conclusions are broad, and authors should also include specific conclusion about the potential probiotic effect of B. subtilis 1554 (it is the main finding obtained from this study).

We thank the Reviewer for this point. We have changed the conclusions accordingly.